# Center-Based Relaxed Learning Against Membership Inference Attacks

**Xingli Fang**[1]                    **Jung-Eun Kim**[*1]

[1]Computer Science, North Carolina State University

## Abstract

Membership inference attacks (MIAs) are currently considered one of the main privacy attack strategies, and their defense mechanisms have also been extensively explored. However, there is still a gap between the existing defense approaches and ideal models in both performance and deployment costs. In particular, we observed that the privacy vulnerability of the model is closely correlated with the gap between the model's data-memorizing ability and generalization ability. To address it, we propose a new architecture-agnostic training paradigm called Center-based Relaxed Learning (CRL), which is adaptive to any classification model and provides privacy preservation by sacrificing a minimal or no loss of model generalizability. We emphasize that CRL can better maintain the model's consistency between member and non-member data. Through extensive experiments on common classification datasets, we empirically show that this approach exhibits comparable performance without requiring additional model capacity or data costs. The code of this work can be found here: `https://github.com/JEKimLab/UAI24_CRL`

## 1 INTRODUCTION

Recently, machine learning has been increasingly questioned with regard to data privacy issues due to the increasing incidents of data leakage in practice. Some studies Fredrikson et al. [2015], Song et al. [2017], Carlini et al. [2019] have addressed that machine learning models tend to memorize the training data, and some techniques are able to even reconstruct those data Salem et al. [2020], Haim et al. [2022]. Research on deploying privacy protection solutions in machine learning models has become a pressing need to play a

better role in privacy-sensitive applications.

In machine learning, Membership Inference Attack (MIA) Shokri et al. [2017] is one of the most important data inference attacks. In membership inference attacks, an attacker tries to determine whether a sample is a member of a target model's training set. MIAs try to develop a proxy to help the attacker distinguish if a sample is a 'member' or 'non-member.' Depending on specific MIAs' policies, the proxy can be a model or a threshold. In general, the attack's difficulty of MIAs depends on the learning task's difficulty of the target model.

We observed that it is the *discrepancy* in the prediction distribution of the model on member data and non-member data that leads to the leakage of privacy. Therefore, we conjecture that if the two prediction distributions coincide, the model will no longer leak membership information. In theory, a perfect model achieves perfect confidence and accuracy in both the training and testing sets. However, it is challenging to train a model that perfectly conforms to the above conception under the current state of the art and resources. Hence, we slightly relax the constraint: while maintaining generalization ability, we would like to develop a model that tries to make the distributions of prediction on the training and testing sets as close as possible.

To achieve this goal, in this paper, we introduce a new training paradigm that can be effective against MIAs by enhancing model generalizability. Our approach is based on the insight that learning models usually show under-confidence and overconfidence in non-member and member data, respectively. Accordingly, our design goal is intuitive: making two distributions close to each other so that the model becomes neither overconfident nor underconfident. Additionally, our approach can easily be applied to train any classification model. In summary, this paper makes the following contributions:

1. We propose `CRL`, a novel defense mechanism, helping classification models gain more robust privacy protection capabilities.

---

*Correspondence

2. To the best of our knowledge, our approach is the first to enhance and maintain models' prediction confidence in nonmember data while mitigating overconfidence in the models' prediction distribution on member data.

3. Through extensive evaluations, we empirically show that our approach outperforms existing defense mechanisms.

## 2 RELATED WORK

### 2.1 MEMBERSHIP INFERENCE ATTACKS AND DEFENSES

MIAs usually attack the target model through a black box model Shokri et al. [2017]. Label-only MIAs Choquette-Choo et al. [2021] can defeat some confidence obfuscation-based methods without confidence score. FAR Rezaei and Liu [2021] was introduced as a MIAs evaluation metric. Song and Mittal [2021] derived a privacy risk score metric for fine-grained privacy analysis and evaluated a series of metric-based attacks. SAMIA Yuan and Zhang [2022] tried to use Gaussian random noise to interfere with the model's reaction. Li et al. [2022] designed a MIAs approach that applies knowledge distillation technology to train shadow models. Adversarial distance MIAs Del Grosso et al. [2022] use Auto attack Croce and Hein [2020] to grab the reaction differences of models.

On the other hand, some studies to defend against MIAs are also proposed. Nasr et al. [2018] proposed a training framework with an inference model to let the target and inference models conduct adversarial regularization. MemGuard Jia et al. [2019] interferes with the prediction distribution of the model by additional noise. Distillation approach for membership privacy (DMP) Shejwalkar and Houmansadr [2021] trains a protected model via selected data and labels from an unprotected model. Kaya and Dumitras [2021] explored when and how data augmentation helps MIAs or defenses while they proposed loss-rank-correlation (LRC) metric to measure the similarity of different augmentation mechanisms' effects on privacy leakage. Exploring how pruning affects neural networks' privacy protection ability, contradictory conclusions were obtained in Yuan and Zhang [2022], Wang et al. [2021]. RelaxLoss Chen et al. [2022] defends the MIAs by relaxing the model's prediction distribution via loss. SELENA Tang et al. [2022] aggregates multiple networks with different training samples for imitating the distribution of the testing set. Yang et al. [2023] designed a reformer to 'purify' the confidence scores. Tan et al. [2023] found there exist trade-offs between parameter size and privacy–utility.

### 2.2 METRIC LEARNING

Wen et al. [2016] proposed a distance metric approach, Center Loss, to learn common features within a class through a learnable class center. Some following studies He et al. [2018], Li et al. [2019], Zhao et al. [2020], Rajoli et al. [2023] improved its performance in the face recognition task. Wang et al. [2019] combined multiple similarity loss functions to achieve better performance. Chen et al. [2020] designed a label-free learning mechanism based on metric learning. SimSiam Chen and He [2021] achieved better accuracy through representation alignment learning under an asymmetric neural network structure while Barlow Twins Zbontar et al. [2021] provided a simpler learning paradigm via cross-correlation matrix. VICReg Bardes et al. [2022a] incorporated the invariance of augmented data, the covariance of the dimensions, and the variance of different samples into the training objectives. And further added the local criterion in Bardes et al. [2022b] . Garrido et al. [2023] extended the two-branch learning paradigm to four-branch learning paradigm via deploying a hypernetwork-based predictor. Fini et al. [2023] combined metric learning techniques in self- and semi-supervised learning to make the model perform better.

## 3 PRELIMINARIES AND PROBLEM FORMULATION

Membership inference attacks aim to detect whether a sample belongs to the target model's training set or not. Hence, it can be formulated as a binary classification task. Suppose there is an attack model $f_a(\cdot; \theta_a)$ with parameters $\theta_a$ and a target model $f(\cdot; \theta)$ with parameters $\theta$. Then the attacker can predict whether the sample $x$ is in or out of the target model's training dataset:

$$\arg\max f_a(f(x; \theta); \theta_a) \qquad (1)$$

If one were to develop an attack model, the model needs to mimic the target model's prediction distribution. A widely adopted solution is the shadow model approach Shokri et al. [2017]. Through some shadow models $f(\cdot; \theta)$ with parameters $\theta_s$, the attack model tries to find the best decision boundary to determine the samples:

$$\max_{\theta_a}[\mathbb{E}_{(x,y)\in D_{in}} f_a(f_s(x; \theta_s); \theta_a) \\ + [\mathbb{E}_{(x,y)\in D_{out}}(1 - f_a(f_s(x; \theta_s); \theta_a))] \qquad (2)$$

where $D_{in}$ is the shadow models' training set and $D_{out}$ is a non-intersection set of the training set. Once the MIA's successful rate is maximized on the shadow models, the attack model can be considered to be successfully trained. In the appendix, a summary and discussion/comparisons of existing defense mechanisms are provided.

## 4 METHODOLOGY

The ideal goal is to make the model privacy-safe with no or little generalizability loss. To achieve the goal, we consider

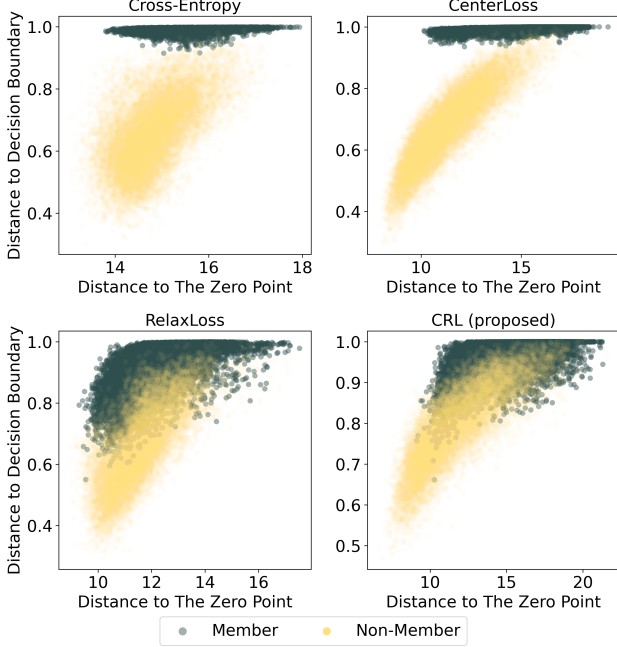

Figure 1: Relationship between the distance to the origin and the distance to the decision boundary in Cross entropy, RelaxLoss, and our proposed approach, CRL. The blue points are Member data while the green points are non-member data. We compute the distance to the decision boundary by subtracting 2nd largest confidence from the 1st largest confidence. Therefore, more overlap between Member and Non-member is better for privacy.

of employing RelaxLoss and CenterLoss. RelaxLoss Chen et al. [2022] can mitigate MIAs by reducing distinguishability between the member and non-member loss distributions. CenterLoss Wen et al. [2016] can enhance the discriminative power of the deeply learned features. The benefits of incorporating both of them are i) it helps to keep the discriminative power of deep features while "relaxing" the model; ii) both of them do not require additional training information or data sources; iii) low training costs and no modifications to the model's inference in the evaluation phase make them easy to apply in most scenarios.

As we describe above, they bring exclusive advantages. However, those advantages are not utilized by simply combining RelaxLoss and CenterLoss. Hence, we propose CRL to utilize their advantages harmoniously.

### 4.1 MECHANISM OF RELAXLOSS

The insight of relaxed loss function (RelaxLoss) Chen et al. [2022] is to adjust the fitting degree of the model on the training set through mini-batches. RelaxLoss sets three stages for achieving this goal. The algorithm can be formulated as

follows:

$$
\begin{aligned}
&\texttt{RelaxLoss}(y, p, \alpha_{rce}, epoch) \\
&= \begin{cases} \mathcal{L}_{ce}, & \text{if } \mathcal{L}_{ce} > \alpha_{rce}, \\ |\mathcal{L}_{ce} - \alpha_{rce}|, & \text{else if } epoch\%2 = 0, \\ \mathcal{L}_{sce}, & \text{otherwise} \end{cases}
\end{aligned} \tag{3}
$$

where $\mathcal{L}_{ce}$ denotes cross-entropy loss function, $\alpha_{rce}$ denotes the threshold hyper-parameter, $y$ denotes the ground-truth, $p$ denotes the prediction probabilities, and $\mathcal{L}_{sce}$ denotes the soft cross-entropy loss function formulated later in Eq. 8 (without logits normalization). RelaxLoss sets a threshold, $\alpha_{rce}$, to judge if the model should become more fitting on the mini-batch sample. When $\mathcal{L}_{ce}$ is below $\alpha_{rce}$, RelaxLoss relieves the fitting degree in two ways. The first case, which only happens in an even number of epochs, reverses the cross-entropy's gradient direction so the model degenerates to the threshold. The other case takes the soft cross-entropy loss function to enable the model to further fine-tune its prediction distribution. The combination of these two cases prevents the model from merely returning to the state before further fitting.

### 4.2 MECHANISM OF CENTER LOSS

As an auxiliary loss function, the center loss Wen et al. [2016] does not act directly on the output layer. Instead, it assumes that a classification model can summarize each class' general features. In detail, each class has a feature vector or a feature map so that the sample is similar to the feature vector or map corresponding to its class after the model processes the sample. To find a set of class centers, we randomly generate a set of vectors, $\{c\}_{i=1}^{C}$, for $C$ classes. Then, we compute the distance between the sample's output of the intermediate layer (usually the second or the global pooling layer) and the corresponding center is,

$$
\mathcal{L}_{ct} = -\frac{1}{|\mathcal{M}|} \sum_{(x,y) \in \mathcal{M}} \frac{1}{2} \| f_e(x; \theta_e) - c_y \|_2^2 \tag{4}
$$

where $\| \cdot \|_2$ is the function of euclidean distance, $\mathcal{M}$ denotes the set of a mini-batch, and $y$ is the ground-truth of the input $x$. And $f_e(\cdot; \theta_e)$ is a model without the classifier part. Representations of samples from the same class are encouraged to approach each other during training. The intuitive idea of why we choose center loss is that the model is more likely to make similar predictions when features in the representation space are closer to their class members. In other words, we try to bring this mechanism to privacy protection so that the member and non-member data of the model overlap more in the representation space, shown in CRL in Fig. 1, to obtain more consistent predictions. However, it does not mitigate the problem that the model is too confident in predicting training samples. We improve it in the next subsection to make it capable of this problem.

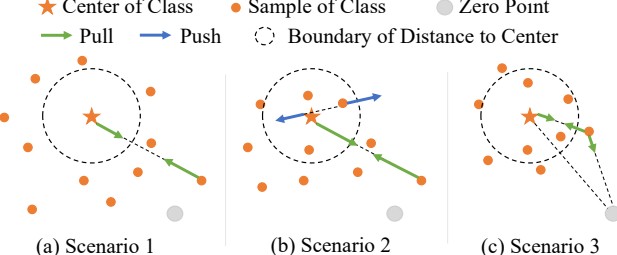

(a) Scenario 1    (b) Scenario 2    (c) Scenario 3

Figure 2: An overview of the proposed relaxed center loss function. The mini-batch determines which scenario to execute.

## 4.3 CENTER-BASED RELAXED LEARNING METHOD (`CRL`)

RelaxLoss shows outstanding performance considering trade-offs between models' generalizability and the prediction distribution gap between training data and testing data, which leads to the behavioral differences of the model on the training members and non-members. However, we observe that the cross-entropy loss function is prone to make the model overconfident in some training samples. To ameliorate the impact of the issue, as a part of our approach, `CRL`, we propose an improved relaxed loss (`ImpRelaxLoss`), which is inspired by Wei et al. [2022]. First, we define the model's function without softmax as $f(\cdot;\theta)$, and the parameters $\theta$ is a superset of $\theta_e$. Then, we revisit the softmax probabilities and define the normalized probabilities:

$$p_i = \frac{e^{g_i}}{\sum_{j=1}^{C} e^{g_j}}, \qquad p_{i,norm} = \frac{e^{g_i/(1+\tau_{rce}\|g\|_2)}}{\sum_{j=1}^{C} e^{g_j/(1+\tau_{rce}\|g\|_2)}} \quad (5)$$

where $g = f(x;\theta)$, and $\tau_{rce}$ is a scaling factor to control the degree of how much the predicting probabilities are normalized. One different point is that we add 1 to $\|g\|_2$ to ensure the denominator is always greater than or equal to 1. Normalization can amplify the loss of difficult samples more so that the model preferentially focuses on difficult samples, intensifying the goal of RelaxLoss. Next, we compute the cross-entropy loss with logit-normalized probabilities:

$$\mathcal{L}_{lce} = -\frac{1}{|\mathcal{M}|} \sum_{(x,y)\in\mathcal{M}} \sum_{i=1}^{C} y_i \log(p_{i,norm}) \quad (6)$$

where $\mathcal{M}$ is the set of a mini-batch, and $p_{i,norm}$ is the normalized probability of the $i$-th class for each sample $y$ in the mini-batch. Afterward, we compute the soft label for the loss function. We change the non-normalized probabilities to produce soft label, $p_{tar}$. The probabilities are averaged except the probabilities of the corresponding class:

$$p_{i,tar} = \begin{cases} p_y, & i = y, \\ (1-p_y)/(C-1), & i \neq y \end{cases} \quad (7)$$

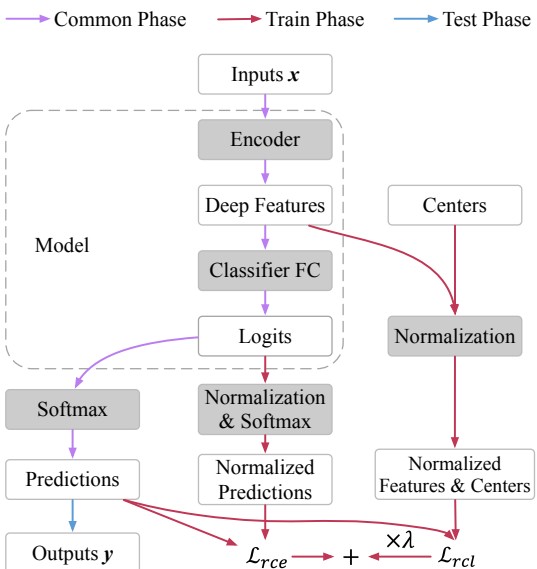

Figure 3: An overview of CRL's training and testing phases. The parameters of centers are a part of the loss function but not a part of the model.

where $p_{i,tar}$ is the $i$-th class probability of the soft label $p_{tar}$. Then, we compute the soft cross-entropy loss as follows:

$$\mathcal{L}_{sce} = -\frac{1}{|\mathcal{M}|} \sum_{(x,y)\in\mathcal{M}} \sum_{i=1}^{C} p_{i,tar} \log(p_{i,norm}) \quad (8)$$

According to the size difference of $\mathcal{L}_{lce}$ and $\alpha_{rce}$, the `ImpRelaxLoss` function is formulated as follows:

$$\begin{aligned} &\texttt{ImpRelaxLoss}(y, p, \alpha_{rce}, \tau_{rcl}, epoch) \\ &= \begin{cases} |\mathcal{L}_{lce} - \alpha_{rce}|, & \text{if epoch}\%2 = 0, \\ \mathcal{L}_{lce}, & \text{else if } \mathcal{L}_{lce} > \alpha_{rce}, \\ \mathcal{L}_{sce}, & \text{otherwise} \end{cases} \end{aligned} \quad (9)$$

Then, `ImpRelaxLoss` is assigned to $L_{rce}$ (Line 20 in Algorithm 1.)

For the next part of `CRL`, we introduce the *Relaxed center loss* function of which overview is described in Fig. 2. Similar to `RelaxLoss`, there are three scenarios in the Relaxed center loss. To determine which scenario to execute, we use the epoch index and distance to the class center (similar to what vanilla center loss does) at the mini-batch level as a metric.

First, we set a distance boundary $\alpha_{rcl}$. The boundary is a hypersphere since we use Euclid distance as a measuring metric. The simplest scenario is shown in Fig. 2 (a): when the average distance of the mini-batch to corresponding centers is larger than the boundary, the centers and samples are driven to close each other. The second scenario illustrated in Fig. 2 (b) occurs only when the index of the current epoch is even. In this scenario, a boundary of distance to centers is

**Algorithm 1** Center-Based Relaxed Learning method (CRL)

**Input**: Training Dataset $\mathcal{D} = \{(x_i, y_i)\}_{i=1}^{N}$ in a random order, training epochs $E$, model learning rates $\tau$, class centers learning rates $\tau_c$, mini-batch size $B$, number of output classes $C$, improved relaxed loss function's threshold value $\alpha_{rce}$, relaxed center loss function's threshold value $\alpha_{rcl}$, normalized factor $\tau_{rce}$ and $\tau_{rcl}$, the joint loss adjustment coefficient $\lambda$;

**Parameter**: Model's encoder part parameters $\theta_e$, classifier part parameters $\theta_c$, class centers' parameter $\{c\}_{i=1}^{C}$;

**Output**: Model $f(\cdot;\theta)$ (inclusive of encoder $f_e(\cdot;\theta_e)$ and classifier $f_c(\cdot;\theta_c)$) with parameters $\theta$ (both $\theta_e$ and $\theta_c$ are inclusive);

1: Randomly initialize the model's parameters $\theta$ and class centers' parameters $\{c\}_{i=1}^{C}$
2: **for** $epoch$ **in** $\{1, 2, \cdots, E\}$ **do**
3:   **repeat**
4:     Sample a mini-batch $\{(x_j, y_j)\}_{j=1}^{B}$ from $\mathcal{D}$
5:     /* Perform forward pass */
6:     $q_j = f_e(x_j; \theta_e), p_j = f_c(q_j; \theta_c)$
7:     $q_{j,norm} = \frac{q_j}{1+\tau_{rcl}\|q_j\|_2}, c_{y_j,norm} = \frac{c_{y_j}}{1+\tau_{rcl}\|c_{y_j}\|_2}$
8:     /* Compute relaxed center loss */
9:     $\mathcal{L}_{ct} = \sum_{j=1}^{B} \|q_{j,norm} - c_{y_j,norm}\|_2^2 / 2B$
10:     **if** $epoch\%2 = 0$ **then**
11:       $\mathcal{L}_{rcl} = |\mathcal{L}_{ct} - \alpha_{rcl}|$
12:     **else if** $\mathcal{L}_{ct} > \alpha_{rcl}$ **then**
13:       $\mathcal{L}_{rcl} = \mathcal{L}_{ct}$
14:     **else**
15:       $t_{j,y} = p_{y_j}$         // confidence of the true class
16:       $t_{j,o} = 1 - p_{y_j}$
17:       $\mathcal{L}_{rcl} = \sum_{j=1}^{B}[t_{j,y}\|q_{j,norm} - c_{y_j,norm}\|_2^2 + t_{j,o}\|q_{j,norm}\|_2^2]/2B$
18:     **end if**
19:     /* Compute improved relaxed loss */
20:     $\mathcal{L}_{rce} = \text{ImpRelaxLoss}(y_j, p_j, \alpha_{rce}, \tau_{rcl}, epoch)$
21:     /* Compute total loss */
22:     $\mathcal{L} = \mathcal{L}_{rce} + \lambda\mathcal{L}_{rcl}$
23:     /* Update model's and centers' parameters */
24:     $c_{y_j} \leftarrow c_{y_j} - \tau_c \nabla \mathcal{L}_{rcl}$
25:     $\theta \leftarrow \theta - \tau \nabla \mathcal{L}$
26:   **until** all training samples are sampled in this $epoch$
27: **end for**

---

defined as a hyperplane in which all points have the same specific distance to the corresponding class centers. It aims to keep the samples around the boundary of distance to their class centers to prevent the collapse of the classifier caused by excessive relaxation when merely relaxing cross entropy. The last scenario shown in Fig. 2 (c) prevents the center and samples from staying too far from the zero point.

*Unlike* RelaxLoss (Cross-Entropy part), relaxed center loss encourages the samples to stay around the connection line between the class center and the origin in this scenario, which implicitly reconstructs the training sample representation's magnitude and direction, helping more relaxed magnitude and narrower angle. In the relaxation process, narrowing the angle helps better generalization Liu et al. [2016]. As

shown in Fig. 1, scenario 3 helps the model's member and non-member samples distribution become sharper than the other two methods. The details are presented in Algorithm 1.

As a result, ImpRelaxLoss and Relaxed center loss compose CRL. The overview of CRL is shown in Fig. 3 and the algorithm is described in Algorithm 1. Through the encoder part of the model, the input gets deep features, then the deep features and their corresponding class centers are normalized. Afterward, the logits produced by the model are also normalized. We then use the logits to compute $\mathcal{L}_{rce}$ and $\mathcal{L}_{rcl}$, respectively. Hyper-parameter $\lambda$ controls the balance between the two losses. Another feature of CRL is that the two components do not execute the corresponding scenarios simultaneously since they have respective thresholds $\alpha_{rl}$ and $\alpha_{rcl}$. In other words, the model can relax the cross entropy loss while maintaining a certain degree of inter-class aggregation in the representation space so that the model can better keep the model's generalizability.

## 5 EXPERIMENTS

### 5.1 EXPERIMENTAL SETTINGS

**Attack and Defense Methods** We mainly evaluate our methods and two state-of-the-art defense methods, AdvReg Nasr et al. [2018] and RelaxLoss Chen et al. [2022]. Also, we evaluate the following common defense methods:

- Label-Smoothing: Guo et al. [2017], Müller et al. [2019]
- Early-Stopping
- Confidence-Penalty: Pereyra et al. [2017]
- DMP: Shejwalkar and Houmansadr [2021]

A baseline cross-entropy approach with no defense mechanism, CE, is also employed against several MIAs:

- Black-Box Attacks: NN-based MIAs (denoted as NN-Based) Nasr et al. [2018], Entropy-based MIAs (denoted as Entropy) Shokri et al. [2017], Modified Entropy-based MIAs (denoted as M-Entropy) Song and Mittal [2021]
- White-Box Attacks: Inputs' Gradient-based MIAs (denoted as Grad-x $\ell_2$) Rezaei and Liu [2021]

For all threshold-based MIAs, we set a threshold for each class to enhance MIAs' successful rate. All these methods require shadow models to mimic the behavior of the target model. When comparing with other defense mechanisms, we use AUC score as an evaluation metric of MIAs to minimize the evaluation biases caused by the different training stability of different approaches. To accommodate the advancement of recent defense approaches, an adaptive attack policy is employed to evaluate defense mechanisms. By an adaptive attack policy, the attacker knows how we train the target

Table 1: Trade-offs between privacy and utility. The MIAs evaluation results are reported in AUC Scores. Higher is better in train/test accuracy (↑) while lower is better in AUC for all MIAs (↓).

(a) On CIFAR-10

| Model | Approach | Train Acc. (%) ↑ | Test Acc. (%) ↑ | NN-Based (%) ↓ | Entropy (%) ↓ | M-Entropy (%) ↓ | Grad-x $\ell_2$ (%) ↓ |
|---|---|---|---|---|---|---|---|
| VGG11 | CE (no defense) | 100.00(±0.00) | 76.46(±0.30) | 76.31(±0.24) | 74.16(±0.26) | 74.90(±0.25) | 75.33(±0.26) |
| | AdvReg | 99.29(±0.31) | 69.52(±0.64) | 72.44(±0.96) | 64.96(±1.12) | 69.23(±1.19) | 69.84(±1.61) |
| | RelaxLoss | 73.00(±3.71) | 64.15(±3.02) | 64.54(±0.77) | 56.89(±0.90) | 60.78(±0.68) | 66.22(±0.67) |
| | CRL (ours) | 89.37(±0.26) | 73.69(±0.36) | 61.33(±0.18) | 61.95(±0.42) | 62.48(±0.40) | 61.95(±0.42) |
| ResNet18 | CE (no defense) | 100.00(±0.00) | 70.31(±0.33) | 88.09(±0.23) | 85.91(±0.32) | 86.44(±0.31) | 86.32(±0.31) |
| | AdvReg | 97.57(±2.00) | 54.97(±5.27) | 77.54(±2.12) | 71.10(±1.07) | 79.28(±1.45) | 70.70(±5.11) |
| | RelaxLoss | 91.56(±1.91) | 69.25(±0.40) | 77.32(±1.33) | 71.51(±2.37) | 72.25(±1.93) | 73.51(±1.69) |
| | CRL (ours) | 86.73(±1.25) | 71.53(±0.50) | 60.21(±1.35) | 63.70(±1.87) | 65.15(±1.90) | 65.34(±1.66) |
| DenseNet121 | CE (no defense) | 100.00(±0.00) | 84.73(±0.33) | 59.00(±0.30) | 65.78(±0.25) | 66.38(±0.23) | N/A |
| | AdvReg | 99.98(±0.02) | 81.72(±0.75) | 55.96(±1.75) | 63.69(±2.81) | 64.94(±2.43) | N/A |
| | RelaxLoss | 92.70(±1.45) | 80.22(±0.94) | 54.38(±0.29) | 57.42(±0.43) | 59.12(±0.34) | N/A |
| | CRL (ours) | 91.82(±0.39) | 83.03(±0.35) | 51.49(±0.06) | 53.28(±0.11) | 55.23(±0.11) | N/A |

(b) On CIFAR-100, data augmentations applied

| Model | Approach | Train Acc. (%) ↑ | Test Acc. (%) ↑ | NN-Based (%) ↓ | Entropy (%) ↓ | M-Entropy (%) ↓ | Grad-x $\ell_2$ (%) ↓ |
|---|---|---|---|---|---|---|---|
| GoogLeNet | CE (no defense) | 99.78(±0.06) | 58.59(±0.32) | 82.70(±0.33) | 77.40(±0.46) | 79.56(±0.38) | 79.78(±0.39) |
| | AdvReg | 99.22(±0.23) | 52.45(±1.22) | 84.02(±1.19) | 76.20(±2.12) | 81.42(±1.29) | 80.07(±1.12) |
| | RelaxLoss | 90.98(±0.76) | 57.90(±0.81) | 65.29(±0.58) | 70.13(±0.73) | 73.79(±0.81) | 74.54(±0.71) |
| | CRL (ours) | 87.39(±0.92) | 58.16(±0.23) | 64.72(±0.61) | 67.10(±0.68) | 70.08(±0.57) | 70.94(±0.47) |
| ResNet18 | CE (no defense) | 100.00(±0.00) | 58.06(±0.62) | 86.88(±0.64) | 82.96(±0.49) | 84.04(±0.45) | 84.20(±0.42) |
| | AdvReg | 99.43(±0.47) | 48.98(±1.21) | 86.99(±1.43) | 79.87(±2.04) | 85.35(±1.59) | 80.03(±0.89) |
| | RelaxLoss | 77.46(±0.33) | 55.28(±0.47) | 69.87(±0.16) | 63.52(±0.16) | 66.60(±0.20) | 69.05(±0.23) |
| | CRL (ours) | 79.74(±0.56) | 57.53(±0.29) | 66.48(±0.31) | 63.80(±0.19) | 65.09(±0.25) | 66.07(±0.25) |
| DenseNet121 | CE (no defense) | 99.01(±0.20) | 62.76(±0.40) | 58.92(±0.36) | 71.34(±0.48) | 74.46(±0.33) | N/A |
| | AdvReg | 99.16(±1.21) | 59.51(±0.80) | 59.90(±2.59) | 73.52(±4.44) | 77.10(±2.13) | N/A |
| | RelaxLoss | 73.46(±0.56) | 58.06(±0.15) | 55.12(±0.25) | 57.03(±0.25) | 61.05(±0.20) | N/A |
| | CRL (ours) | 77.39(±0.59) | 60.32(±0.50) | 50.23(±0.23) | 59.98(±0.27) | 61.58(±0.25) | N/A |

(c) On SVHN

| Model | Approach | Train Acc. (%) ↑ | Test Acc. (%) ↑ | NN-Based (%) ↓ | Entropy (%) ↓ | M-Entropy (%) ↓ | Grad-x $\ell_2$ (%) ↓ |
|---|---|---|---|---|---|---|---|
| VGG11 | CE (no defense) | 99.98(±0.00) | 92.66(±0.04) | 53.09(±0.02) | 54.14(±0.06) | 54.63(±0.07) | 54.51(±0.06) |
| | AdvReg | 98.72(±1.25) | 90.72(±1.63) | 52.40(±0.94) | 54.53(±1.38) | 55.31(±1.49) | 54.54(±0.87) |
| | RelaxLoss | 95.42(±0.15) | 91.65(±0.04) | 51.67(±0.09) | 52.02(±0.11) | 52.34(±0.10) | 52.36(±0.10) |
| | CRL (ours) | 94.70(±0.53) | 91.50(±0.55) | 50.15(±0.15) | 52.32(±0.16) | 52.48(±0.20) | 52.38(±0.18) |
| ResNet18 | CE (no defense) | 100.00(±0.00) | 93.04(±0.07) | 52.93(±0.14) | 54.74(±0.10) | 55.07(±0.09) | 54.79(±0.09) |
| | AdvReg | 99.90(±0.17) | 90.47(±1.37) | 52.18(±0.51) | 55.14(±0.37) | 55.79(±0.47) | 55.07(±0.13) |
| | RelaxLoss | 95.39(±0.21) | 93.07(±0.25) | 51.80(±0.06) | 52.12(±0.05) | 52.23(±0.08) | 52.08(±0.05) |
| | CRL (ours) | 95.82(±0.24) | 93.13(±0.27) | 50.04(±0.04) | 51.92(±0.06) | 52.02(±0.07) | 51.89(±0.07) |

model and so applies the same way to the shadow models. In NN-based MIAs, we train five shadow models to train an attack model. The other MIAs, which are threshold-based, do not require shadow models since the AUC score is to evaluate the target model and its training and testing datasets directly use corresponding metrics.

**Datasets** We evaluate our methods on CIFAR-10, CIFAR-100 Krizhevsky et al. [2009], SVHN Netzer et al. [2011], and ArXiv-10 (NLP classification dataset) Farhangi et al. [2022]. Their settings are introduced in detail in the Appendix. In particular, in CIFAR-100, common **data augmentation** techniques are applied. To guarantee that target models and shadow models are trained in datasets without intersections, we split the whole dataset into target set and

shadow set, with the ratio 0.5 : 0.5. Then, we evenly split dataset into training and testing sets. To utilize limited data, we use increasing random seeds to ensure that shadow models are trained on different training sets. For reproducibility, we set the default random seed to 0.

**Models** In CIFAR-10, we evaluate our approach and other related methods with VGG11 Simonyan and Zisserman [2015] with batch normalization layers, ResNet18 He et al. [2016], and DenseNet121 Huang et al. [2017]. In CIFAR-100, their performances are evaluated by GoogLeNet Szegedy et al. [2015], Resnet18, and DenseNet121. In SVHN, we apply VGG11 and ResNet18. In AdvReg, we follow their settings to produce the inference attack model. In ArXiv-10, hierarchical attention network (HAN) Yang et al. [2016] is

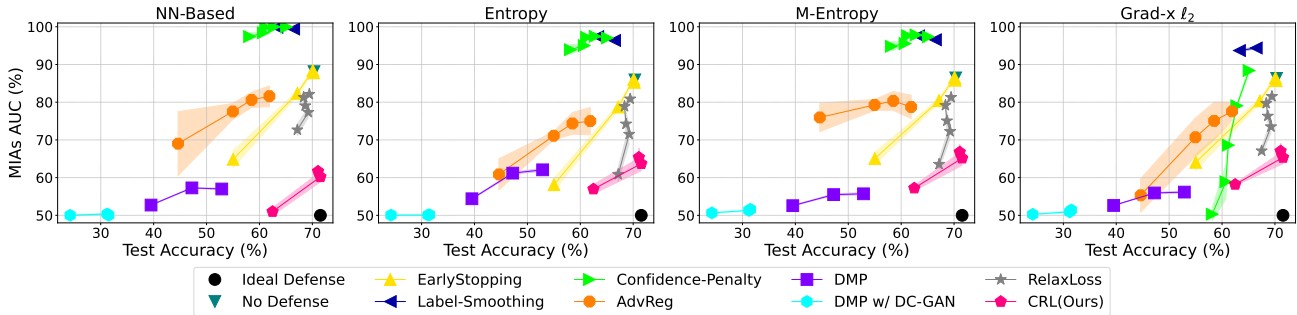

Figure 4: Performance of defenses against adaptive attacks (ResNet18, CIFAR-10).

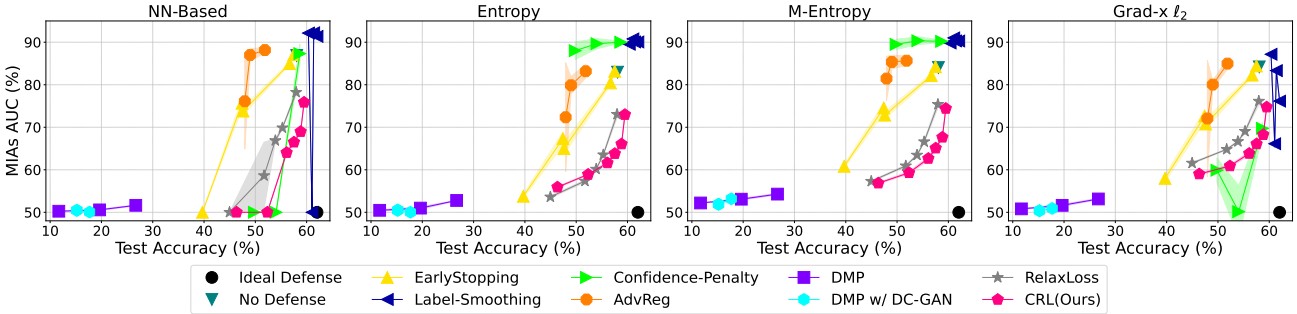

Figure 5: Performance of defenses against adaptive attacks (ResNet18, CIFAR-100, data augmentation applied).

used for evaluation. For the attack model in NN MIAs, we use a 4-layer fully connected neural network with hidden layer sizes [128, 64]. We also apply ReLU Agarap [2018] and dropout Srivastava et al. [2014] to it. Because the cost of the white-box attack on DenseNet is too high, we only report the results of other models under the white-box attack in this paper.

**Configurations** We apply stochastic gradient descent (SGD) optimizer to train all models except the inference attack models in AdvReg approach (the Adam optimizer Kingma and Ba [2014] employed). To extract deep features, we generally choose the last global average pooling layer or the 2nd last fully connected layer. The learning rate of the class centers is constantly set at 0.001. To keep a consistent mini-batch size across all GPUs used for training, we set the mini-batch size at 32 when we train DenseNet121, and 256 for the other neural networks. For training attack models in NN MIAs, we always set the mini-batch size at 256. Unless otherwise stated, all experiments in the experiments section are repeated in five independent runs. The variance of our method's results is insignificant as the maximum variance is under 3.5%.

## 5.2 COMPARISON AND ANALYSIS

**On CIFAR-10** As shown in Table 1a, our approach performs better in terms of trade-offs of testing accuracy and privacy-preserving across all three neural networks. An ex-

pected result is that different models have different generalization capabilities, leading to gaps in their natural privacy-preserving abilities. Another discovery is that the model has the upper limit of representation ability since computation capacity, which depends on the depth, width, and computation functions, is positively correlated with privacy protection capability.

We further evaluate more approaches on CIFAR-10 shown in Fig. 4. We validate that our method can better mitigate privacy leakage without losing generalizability. In the figure, although Label-Smoothing is a bit better at testing accuracy, it cannot help privacy preservation. Earlystopping is not as good as our method and Relaxloss because it cannot determine which samples should be relaxed or further learned. DMP does not perform satisfactorily through both splitting and synthesized data. This is because splitting the training set can hurt accuracy, and it is impossible to establish a strong GAN with non-excessive data. Besides, knowledge distillation can still leak the original model's privacy via non-training or even OOD data enquires Nayak et al. [2021]. The noteworthy point is that our method and RelaxLoss are more effective on models with more computational capacity. This exhibits more significant results on CIFAR-100.

**On CIFAR-100** In Table 1b, a trend that our approach can alleviate the predictions overconfidence on the training set more while keeping prediction confidence on the testing set more significant. Among all, our approach shows the most superior privacy-protection capacity in NN, Entropy,

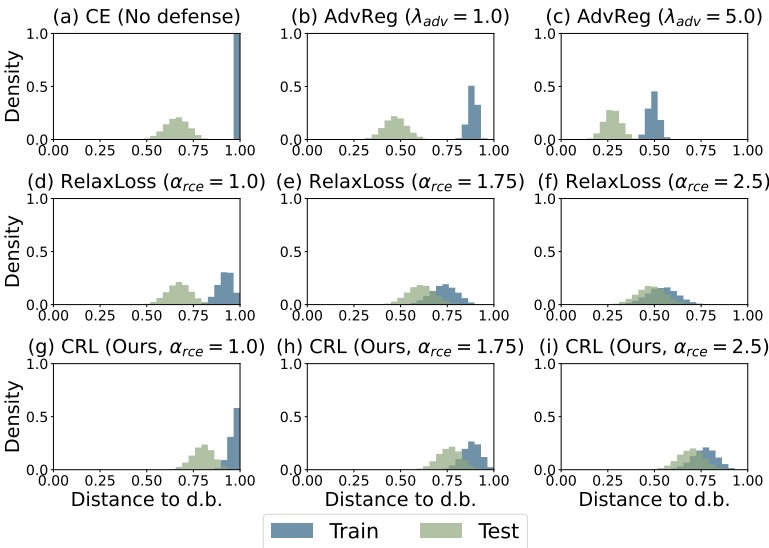

Figure 6: Histograms of distance to decision boundary on CIFAR-100 with ResNet18 trained and tested in CE (no defense), AdvReg, RelaxLoss and our approach with various hyper-parameter settings.

M-Entropy and Grad-x $\ell_2$ MIAs. We empirically found that AdvReg is unsuitable for simultaneous deployment with data augmentation to train models. Under data augmentation, it becomes more difficult for the model to maintain accuracy in adversarial regularization training. In particular, on ResNet18, AdvReg respectively exhibits even 0.11% and 1.29% increase in NN-Based and M-Entropy MIAs while there is about 10% decrease in test accuracy. Also, it has a little impact on Entropy and Grad-x $\ell_2$ MIAs. Through experiments in the data augmentation scenario, we found that the AdvReg model could gradually gain privacy protection capabilities only after the loss of test accuracy reaches a certain magnitude.

We further explore how the testing accuracy and MIAs accuracy change when we enhance the privacy-related hyper-parameter settings. In Fig. 5, we explored the trends between testing accuracy and MIAs AUC scores in more defense approaches and settings. Our approach always achieves more privacy preservation with less testing accuracy loss. Under a more challenging situation (fewer samples per class and harder task difficulty), DMP performs poorer than when it is in CIFAR10. RelaxLoss also shows a stable trend of trade-offs between testing accuracy and privacy-preserving. In our method and the other two defense methods, AdvReg shows the lowest testing accuracy when the three defenses are at the same privacy level. One of the main reasons for this phenomenon is that AdvReg requires separating a part of the training set as a conference set, resulting in additional data cost and the model's generalizability pays for that. Besides AdvReg, Confidence-Penalty approach also shows the effectiveness on Grad-x $\ell_2$. This is because both methods reduce the true class prediction probability, which effectively combats the cross-entropy loss function.

As shown in Fig. 6, we experimented on ResNet18 using three approaches with different levels of privacy settings. We found that the prediction distribution of the model without privacy-preserving measures on the training set is significantly different from that on testing. Also, the three defense approaches show distinctive differences. AdvReg's testing prediction distribution shifts to the decision boundary farthest among all nine charts, lending to more losses of the model's utilities. However, there is still a clear distribution gap between the training and testing sets, which makes it not as effective as RelaxLoss and our approach. As for RelaxLoss, the area of overlap between the two distributions has been significantly improved.

Compared to RelaxLoss, CRL shows two advantages: (i) it enhances testing confidence while alleviating training over-confidence. (ii) it maintains the testing confidence distribution better. The first advantage helps the model achieve better testing accuracy. The second helps training and testing distributions overlap at an earlier stage. All three charts of CRL show better testing confidence than the others. Even with a large overlap such as Fig. 6i, our method is still more confident in the testing set than the model with no defense.

**On SVHN** Shown in Table. 1c, different from CIFAR datasets, both VGG11 and ResNet18 show over 90% testing accuracy, which means a smaller distribution gap between testing predictions and training predictions than the other two datasets. Overall, we evaluate our method on datasets of different difficulties. In ResNet18, our approach outperforms in NN-Based MIAs defense and exhibits comparable results in other MIAs. However, although the data in this dataset is sufficient to achieve excellent testing accuracy, AdvReg suffers from a significant testing accuracy decrease without

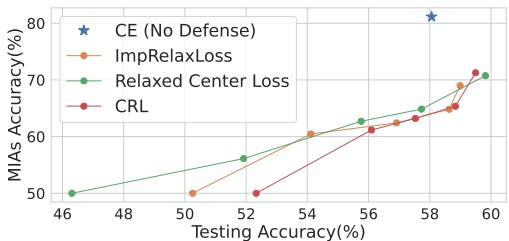

Figure 7: Ablation study of defenses with different components (ResNet18, CIFAR-100).

Table 2: Ablation components.

| Approach | Relaxed Policy | Centers | Normalization |
|---|---|---|---|
| ImpRelaxLoss | ✓ | ✗ | ✓ |
| Relaxed Center Loss | ✓ | ✓ | ✗ |
| CRL | ✓ | ✓ | ✓ |

enhancement of privacy. This also reflects that methods requiring additional data can lead a model to data starvation. On both VGG11 and ResNet18, our method performs similarly to `RelaxLoss`, further suggesting that large models have more potential for privacy protection.

**ArXiv-10**   Shown in Table 3, CRL has a better defense effect against the three kinds of MIAs. With such privacy preservation, RelaxLoss and CRL are able to maintain more comparable testing accuracy, as validated in Fig. 8.

## 5.3   ABLATION STUDY

Here, we evaluate how the main components affect our approach. We evaluate the importance of two components that compose `CRL`: (i) `ImpRelaxLoss` and (ii) Relaxed center loss. The details of the components are described in Table 2. Here, we do not apply normalization when experimenting with Relaxed center loss to avoid the impact of normalization. As shown in Fig. 7, we apply all three approaches to train ResNet18 on CIFAR-100. Pure Relaxed center loss can help Resnet18 achieve the highest testing accuracy. However, as hyper-parameters $\alpha_{rce}$ and $\alpha_{rcl}$ increase, which means more privacy protection, it gradually degenerates to a level comparable to `RelaxLoss`. `ImpRelaxLoss` looks quite different. Overall, it is slightly but more effective than relaxed center loss, especially when they defend the MIAs completely. However, it falls short compared to Relaxed center loss in the highest testing accuracy. Finally, `CRL`, combining all components, can achieve the best performance in most situations. Normalization further enhances the relaxed center loss, allowing the model to resist MIAs with about 6% higher testing accuracy. One of the main reasons is that normalization increases the loss with insufficient confidence, making the model pay more attention to those samples.

Table 3: Trade-offs between privacy and utility for ArXiv-10 on HAN. The MIAs evaluation results are reported in AUC Scores.

| Approach | NN-Based (%) ↓ | Entropy (%) ↓ | M-Entropy (%) ↓ |
|---|---|---|---|
| CE (no defense) | 54.66(±0.16) | 51.24(±0.11) | 54.94(±0.24) |
| AdvReg | 54.83(±0.14) | 51.08(±0.18) | 54.88(±0.29) |
| RelaxLoss | 50.38(±0.05) | 51.13(±0.14) | 54.80(±0.16) |
| CRL (ours) | 50.46(±0.07) | 51.07(±0.13) | 53.60(±0.15) |

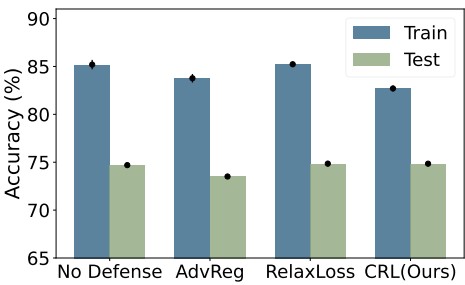

Figure 8: Performance of different approaches on ArXiv-10.

## 6   DISCUSSION AND LIMITATIONS

Compared with RelaxLoss, CRL is more effective for the model to maintain better generalizability while pursuing membership privacy. However, there are still room for improvement. On the one hand, it is still difficult for CRL to make the distribution of prediction of the model on the testing and the training sets fully consistent without any loss of models' utilities - thus that is a tradeoff. On the other hand, more hyper-parameters increase the complexity of the search space. Besides, the performance gap between CRL and Relaxloss will gradually become narrower as the model accuracy decreases a lot. We hope this work is inspiring so that in the future more research efforts can be contributed to improve the solution.

## 7   CONCLUSION

In this paper, we present `CRL`, an easy-deployed yet effective training paradigm that is able to help classification models defend against privacy attacks with minimal or no loss, and even with the improvement of models' generalization abilities. It makes the model behave *indistinguishably* on member and non-member data by encouraging the membership prediction distribution and the non-membership distribution as *consistent* as possible. Our experiments show the outperforming results compared to the well-known defense approaches and the state-of-the-art approaches.

## ACKNOWLEDGEMENT

This publication is supported by the National Science Foundation under Grant no. 2302610. Any opinions, findings, and conclusions or recommendations expressed in this material are those of the author(s) and do not necessarily reflect the views of the National Science Foundation.

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

# SUPPLEMENTARY MATERIAL

## A   EXISTING COMPARABLE DEFENSE METHODS

**Adversarial Regularization (`AdvReg`)**   The adversarial regularization Nasr et al. [2018] method introduced an inference attack model to the training framework. With a part of the training data as a conference set, a target model is encouraged to learn the classification target and try to deceive the inference attack model. In contrast, the inference attack model tries to attack the target model. *Compared with our methods*, it requires splitting the training set, leading to additional data costs. Another weakness is that the model is prone to collapse in the adversarial training process, leading to a significant decrease in accuracy or complete degeneration. Our experiment results also show this aspect.

**Distillation for Membership Privacy (`DMP`)**   Distillation for membership privacy Shejwalkar and Houmansadr [2021] develops a meta-regularization technique based on knowledge transfer. The key difference between AdvReg and DMP is that DMP sets up an entropy-based criterion to produce the reference set via selecting decision-boundary-unimpactful samples. *Compared with our methods*, CRL can achieve a similar effect directly by adjusting the prediction distribution of the model on the training set, avoiding the potential accuracy loss caused by distilling data.

**Relaxed Loss (`RelaxLoss`)**   Relaxed loss Chen et al. [2022] tries to limit the loss of mini-batches to a fixed value near a fixed value $\alpha_{rce}$. It aims to solve cross-entropy's privacy problem that the model always overfits the training data set. Through the improvement of traditional cross-entropy (CE) loss, RelaxLoss is divided into three stages: (i) normal cross-entropy loss, (ii) keeping loss, and (iii) target dispersion. When a mini-batch's average loss is greater than $\alpha_{rce}$, it executes normal CE loss. When the loss is less than $\alpha_{rce}$ and the index of the current epoch is even, the absolute value sign reverses the direction of the gradient. When the index is not even, hard labels are replaced with soft labels produced by predictions. According to the above method, RelaxLoss can limit the loss of the samples in the training set close to a preset value so as to mitigate the models' overconfidence. *For our method*, we additionally ensure that samples in each class share some common feature representations, which is beneficial to improve and keep the model's generalizability.

**Early-Stopping**   The early-stopping method aims to end the training earlier to make the model fit training data less. However, relax policy in both `CRL` and `RelaxLoss` prevents samples that were first fitted from being further fitted and continues fitting the rest of the samples, always leading to better performance than early-stopping unless the model is heavily over-fitting.

**DP-SGD**   `DP-SGD` mixes noise into the classifier during training to provide a reliable privacy guarantee. However, keeping with both acceptable generalization ability loss and privacy guarantees is still challenging Jayaraman and Evans [2019]. Compared with `CRL`, `CRL` gives a more appropriate and specific solution based on the fitting degree of samples.

## B   MORE DETAILS IN EXPERIMENTAL SETTING

**Dataset**   CIFAR-10 and CIFAR-100 are popular image classification datasets, which consist of $60,000$ color images with the size of $32 \times 32$. SVHN, a digits classification dataset, with 10 classes for digits from 0 to 9 and $32 \times 32$ image size, includes over $600,000$ digit images in natural scenes. As for both CIFAR datasets, we apply the commonly used data normalization to the original training and testing sets. In CIFAR-100, data augmentation techniques, random cropping and random flipping, are applied to enhance the model's generalization ability. As for SVHN, we use the matrices of the original RGB image as inputs.