# OpenReview forum: "Center-Based Relaxed Learning Against Membership Inference Attacks"
_auai.org/UAI/2024/Conference — UAI 2024 poster_

### Official Review · Reviewer_WuCt · 2024-03-13

**Q2-1 Originality-Novelty:** 2
**Q2-2 Correctness-Technical Quality:** 3
**Q2-5 Clarity Of Writing:** 3

**Q1 Summary And Contributions:**

This paper presents a training loss CRL that encourages consistency in model's predicted distribution on member and non-member data. CRL is proposed for defending against membership inference attacks.

**Q2-3 Extent To Which Claims Are Supported By Evidence:**

3: Good: the main claims are supported by convincing evidence (in the form of adequate experimental evaluation, proofs, (pseudo-)code, references, assumptions).

**Q2-4 Reproducibility:**

3: Good: key resources (e.g. proofs, code, data) are available and key details (e.g. proofs, experimental setup) are sufficiently well-described for competent researchers to confidently reproduce the main results.

**Q3 Main Strengths:**

- The paper is well-organized.
- Decisions in methodology are mostly well-explained with summarized intuition behind each decision.
- There are extensive experimental results to support the claims.

**Q4 Main Weakness:**

- Section 4 lacks high-level intuition behind choosing the relaxloss and center loss to incorporate in CRL.
- Some visual results lack statistical explanations.
- Experiments are conducted on datasets with same dimension.

**Q5 Detailed Comments To The Authors:**

- In section 4, what is the high-level intuition behind choosing relaxloss and center loss? Do they each bring a mutually-exclusive advantage or do they serve towards the same goal?
- How are the hyper parameter such as $\alpha_{rce}$ chosen? What are the intuitions behind choice of their values?
- In section 4.2, "To find a set of class centers, we randomly generate s set of vectors ...", why are the vectors randomly generated? Does the choice of these vectors affect the performance of model?
- In figure 1, data for member is hidden behind data for non-member.
- What is the intuition behind "preventing samples from staying too far from the zero point". Why is this important for models.
- In results such as those in figure 1 and figure 6, are there statistical results to show the difference between the distributions?
- All three datasets are for 32x32 images, any experiments on higher resolution data?
- In figure 5, some lines are not monotonic, especially label smoothing on CIFAR100, what is the intuition behind this?

**Q9 Complying With Reviewing Instructions:**

Yes

---

> ### Author Rebuttal · Authors · 2024-04-08
>
> **Rebuttal By Authors (1/2)**
>
> We thank the reviewer for the constructive comments. We hope that we were able to address all your concerns. We sincerely hope the responses help you consider raising your score.
>
> > [1] In section 4, what is the high-level intuition behind choosing relaxloss and center loss? Do they each bring a mutually-exclusive advantage or do they serve towards the same goal?
> Thank you for your question.
>
> (1). The ideal goal is to make the model privacy-safe without generalizability loss. RelaxLoss can mitigate MIAs by reducing distinguishability between the member and non-member loss distributions. CenterLoss can enhance the discriminative power of the deeply learned features. We provide a new updated chart with CenterLoss for Fig 1, and we kindly ask you to refer to it – please click [HERE-NewPlots](https://anonymous.4open.science/r/UAI_CRL-269/CRL_UAI24Rebuttal.pdf). There are reasons why we chose them: i) our main high-level intuition is to keep the discriminative power of deep features while “relaxing” the model. ii) both of them do not require additional training information or data sources. iii) Low training costs and no modifications to the model's inference in the evaluation phase make them easy to apply in most scenarios.
>
> (2). As we describe above, they bring exclusive advantages. However, those advantages are not utilized by simply combining RelaxLoss and CenterLoss. That is the reason why we proposed CRL to exploit the advantages of them.
>
> > [2] How are the hyper parameter such as α_rce chosen? What are the intuitions behind choice of their values?
>
> Thank you for your question. α_rce is mainly affected by the number of classes. That is because the initial model’s loss distribution on the dataset varies when the number of classes changes. We did not know its best combination before we did initial experiments on the specific dataset. Hence we ran extensive experiments and empirically chose it based on such observations. As for the choice of the values, for the same model on the same dataset, when we increase the α_rce, more privacy can be protected while more testing accuracy could be lost.
>
> > [3] In section 4.2, "To find a set of class centers, we randomly generate s set of vectors ...", why are the vectors randomly generated? Does the choice of these vectors affect the performance of model?
>
> Thank you for your question. The reason why we generate centers randomly is that models trained with cross-entropy typically exhibit an actinomorphous distribution of its representation features on the training set (which means there is no single class center). This kind of distribution can be found in existing studies such as [2]. In other words, centers are learned rather than being “born.” A random choice does not affect the performance of the model unless they are set to a very unique value.
>
> > [4] In figure 1, data for member is hidden behind data for non-member.
>
> We understand your concern. We already noticed the issue The issue was because there are  more than 20K data points in the chart. Hence, although we used translucent colors, the green dots overwritten the blue dots because there are too many. To visualize the overlap, we have them in two ways (one with Member and the other with Non-Member in the front) . We kindly ask you to refer to the new charts – please click [HERE-NewPlots](https://anonymous.4open.science/r/UAI_CRL-269/CRL_UAI24Rebuttal.pdf). We will update them in the revised manuscript so that the overlap can be visible.
>
> > [5] What is the intuition behind "preventing samples from staying too far from the zero point". Why is this important for models.
>
> We observed that the testing samples’ representation feature is more likely to have a shorter distance to the zero point. Therefore, one of the aims is to make training samples and testing samples overlap more. The other aim is that it can prevent the samples from being pushed to the decision boundary. This is reflected in the third formula in (3) of RelaxLoss.
>
> > [6] In results such as those in figure 1 and figure 6, are there statistical results to show the difference between the distributions?
>
> Yes, that is correct. They show that the discrepancy in prediction distributions between training and testing sets is mitigated by privacy-preserving approaches. In Fig.1, CRL clearly exhibits a high degree of overlap between the training and testing distributions. In Fig.6, not only does CRL mitigate distinguishability between the member and non-member prediction distributions, but it also keeps the discriminative capability of the deeply learned features (it shows improvement of the prediction distribution on the testing set.)

---

### Official Review · Reviewer_3iwF · 2024-03-21

**Q2-1 Originality-Novelty:** 3
**Q2-2 Correctness-Technical Quality:** 3
**Q2-5 Clarity Of Writing:** 3

**Q1 Summary And Contributions:**

The authors propose CRL a training paradigm that represents a countermeasure against Membership Inference Attack. The goal of CRL is to make the model behave indistinguishably on member and non-member data by encouraging the membership prediction distribution and the non-membership distribution to be more consistent. They experimented against multiple SOTA approaches on image-based classification tasks.

**Q2-3 Extent To Which Claims Are Supported By Evidence:**

2: Fair: the main claims are somewhat supported by evidence (but the experimental evaluation may be weak, or does not match entirely with the claims, important baselines may be missing, proofs contain important ideas but lack rigor, algorithmic details are only discussed superficially, references are imprecise, assumptions are not sufficiently motivated or explicated, etc.).

**Q2-4 Reproducibility:**

4: Excellent: key resources (e.g. proofs, code, data) are available and key details (e.g. proof sketches, experimental setup) are comprehensively described for competent researchers to confidently and easily reproduce the main results.

**Q3 Main Strengths:**

- Straight to the point and well-written
- Direct improvement over SOTA (Good empirical results)

**Q4 Main Weakness:**

- Incremental work with few novelties
- No formal analysis
- Empirically tested only on image-based use cases.

**Q5 Detailed Comments To The Authors:**

- I may have missed it in the paper. But are the introduced losses used exclusively for the training? Or are the centers and normalized prediction still used in testing time? (to maybe limit the decrease in generalization).
I think it should be crystal clear in the paper. Because it raises another question:  Are the parameters such as "Centers" shared with the attacker? If yes could they be leveraged by an attacker?


- One main problem with the paper is the lack of formal analysis on why this new loss helps the distribution of losses (non-member vs member) be closer together.


- Since most of the advantages of CRL are shown through empirical analysis. It would have been better to have other use cases different than the one of the images. Maybe one voice, text, tabular...etc.
Also, it misses a user case with a high number of parameters and data.

- It should have been made more explicit as to why DP-SGD was not discussed much in the paper. I would guess that as a direct competitor of CRL, the privacy/utility trade-off would be too bad. But can they be combined? will it be useful?
This needs to be discussed in the paper.

**Q9 Complying With Reviewing Instructions:**

Yes

---

> ### Author Rebuttal · Authors · 2024-04-08
>
> **Rebuttal by Authors (1/2)**
>
> We thank the reviewer for the thoughtful comments. We hope that we were able to address all your concerns. We sincerely hope the responses help you consider raising your score.
>
> > [1] Are the introduced losses used exclusively for the training? Or are the centers and normalized prediction still used in testing time? (to maybe limit the decrease in generalization). I think it should be crystal clear in the paper. Because it raises another question: Are the parameters such as "Centers" shared with the attacker? If yes could they be leveraged by an attacker?
>
> Thank you for your comment. We would like to kindly ask you to refer to Fig.3. The CRL is applied in the training phase. In the testing phase, the model acts in the common way. The paramters of centers are not shared with the attacker since it is a part of the loss function rather than a part of the model. When we do an adaptive attack, we assume that the attacker knows how the model is trained so that the attacker can imitate it.  We appreciate your suggestion, and we will modify the corresponding texts to describe them more clearly.
>
> > [2] One main problem with the paper is the lack of formal analysis on why this new loss helps the distribution of losses (non-member vs member) be closer together.
>
> We are happy to explain it. The disparity of the model’s recreation between training samples and testing samples leads to the success of the membership inference attack. In other words, if they are in the same distribution, the privacy risk can be solved or at least highly mitigated. As for CRL, first, in Fig.1, we explore what RelaxLoss is not capable of and understand that there is room for RelaxLoss to improve. Then, we introduce CenterLoss which improves Cross-Entropy to mitigate the issue. Then, in Sec. 4.3., we propose CRL, which we build on top of the advantages of Improved RelaxLoss and Relaxed Center Loss, but we advance CRL with our novel observation, which is devised from the discrepancy of training and testing distributions. In short, the rationale of CRL is to keep the discriminative power of deep features (CenterLoss) while “relaxing” the model (RelaxLoss). We understand your concern and will elaborate on it in the revised manuscript.
>
> > [3] Since most of the advantages of CRL are shown through empirical analysis. It would have been better to have other use cases different than the one of the images. Maybe one voice, text, tabular...etc. Also, it misses a user case with a high number of parameters and data.
>
> Thank you for your suggestion. To further show our approach’s applicability and scalability, we have run our approach on a text classification dataset, ArXiv-10 [1] with the Hierarchical attention network [2]. Here are the results averaged by five independent runs, and the standard deviations are parenthesized:
> | Approach   | Testing Acc. % | NN-based (AUC) % | Entr. (AUC) % | Mentr. (AUC) % |
> |------------|----------------|------------------|---------------|----------------|
> | No Defense | 74.69(±0.16)   | 54.66(±0.16)     | 51.24(±0.11)  | 54.94(±0.24)   |
> | Advreg     | 73.51(±0.20)   | 54.83(±0.14)     | 51.08(±0.18)  | 54.88(±0.29)   |
> | RelaxLoss  | 74.85(±0.23)   | 50.38(±0.05)     | 51.13(±0.14)  | 54.80(±0.16)   |
> | Ours       | 74.84(±0.19)   | 50.46(±0.07)     | 51.07(±0.13)  | 53.60(±0.15)   |
>
> We are going to include any further details of the comparison in the revised manuscript. As for higher parameters, the trend of the results is mainly determined by the architecture of the model and the training difficulty of the dataset. Models with the same architecture and different sizes can often obtain similar trends (unless one’s capacity is extremely small).

---

### Official Review · Reviewer_jx49 · 2024-03-22

**Q2-1 Originality-Novelty:** 3
**Q2-2 Correctness-Technical Quality:** 3
**Q2-5 Clarity Of Writing:** 4

**Q1 Summary And Contributions:**

The paper focuses a model agnostic defense against membership inference attacks (MIAs). The premise of the paper follows from the discrepancy in the prediction distribution on member (training data) and non-member data (e.g., test data). The authors propose a novel approach that attempts to bring the member and non-member prediction distributions closers to one another while making the model neither underconfident nor overconfident using an amalgam of relax loss and center loss. The empirical evaluation demonstrates how the proposed method CRL, center-based relaxed learning, outperforms existing methods against MIAs.

**Q2-3 Extent To Which Claims Are Supported By Evidence:**

4: Excellent: all claims are supported by very convincing evidence (in the form of comprehensive experimental evaluation, rigorous mathematical proofs, detailed (pseudo-)code, precise references, well-motivated and realistic assumptions) and the authors deliver what they promise.

**Q2-4 Reproducibility:**

4: Excellent: key resources (e.g. proofs, code, data) are available and key details (e.g. proof sketches, experimental setup) are comprehensively described for competent researchers to confidently and easily reproduce the main results.

**Q3 Main Strengths:**

1. The paper is well written and concise. The authors cover sufficient preliminary information and present their ideas using multiple modes, including figures, tables, and supporting text, for clarity. I enjoyed reading the paper.
1. The empirical evaluation is complete and robust. The paper covers multiple SOTA attack and defense methods, datasets, and models. The discoveries drawn are interesting and offer new insights.
1. The paper has potential for high impact. There is limited work in MIAs that focuses on lowering the discrepancy in prediction distributions between member and non-member data. This has strong implications for MIAs and opens up a new directions of research.
1. The paper results are easily reproducible based on the information provided.

**Q4 Main Weakness:**

1. The technicalities of the proposed approach are not entirely novel. It is a somewhat straightforward amalgamation of two existing methods.
1. The intuition for why the amalgamation of RelaxLoss and CenterLoss is useful in reducing prediction distributions is not clear. This could be improved with more clear explanations (see detailed comments (1) and (2)).
1. While the experiments are thorough, the proposed method has comparatively different impact on different models. This is highlighted a bit in Section 5.2. More intuition as to why this occurs may further help strengthen the motivation for using CRL.

**Q5 Detailed Comments To The Authors:**

1. There are some minor spots which could aid from more careful explanation. Particularly, Sections 4.1 and 4.2 came out of surprise. It would be useful to add a small summary at the beginning of Section 4 and introduce RelaxLoss and Center Loss there before diving into them deeper.
1. The description of RelaxLoss is a little confusing. What does "more fitting" mean? Using that terminology also leads to questions about overfitting. Please refine the description of RelaxLoss (e.g., focus on it allowing flexibility and some degree of uncertainty, which also prevents overfitting) and describe the impact of the threshold value (lower = ?, higher = ?).
1. It is unclear what Figure 1 is trying to tell us. There are brief descriptions in Sections 4.2 and 4.3. However, it would be beneficial to add more text explaining it, especially since it represents the core intuition of the work.
1. It would be interesting to see CE with Center Loss in Figure 1.
1. Where is T_rcl used in equation 9?
1. Understanding Figure 2 required a lot of back and forth between the text, equation 9, and Algorithm 1. Adding more description in the caption of Figure 2 and/or renaming "Scenario X" to something more meaningful may help.
1. The abstract mentions lack of consideration of deployment cost as a gap in the field. However, there is no evaluation of the deployment cost of CRL. Adding some results on cost will strengthen the paper.

**Q9 Complying With Reviewing Instructions:**

Yes

---

> ### Author Rebuttal · Authors · 2024-04-08
>
> We thank the reviewer for the effort and time to review out paper. We hope that we were able to address all your concerns.
>
> > [1] There are some minor spots which could aid from more careful explanation. Particularly, Sections 4.1 and 4.2 came out of surprise. It would be useful to add a small summary at the beginning of Section 4 and introduce RelaxLoss and Center Loss there before diving into them deeper.
>
> Thank you for pointing it out. We agree with you. We will provide an overview at the beginning of Section 4 to provide a brief roadmap of the section for the readers.
>
> > [2] The description of RelaxLoss is a little confusing. What does "more fitting" mean? Using that terminology also leads to questions about overfitting. Please refine the description of RelaxLoss (e.g., focus on it allowing flexibility and some degree of uncertainty, which also prevents overfitting) and describe the impact of the threshold value (lower = ?, higher = ?).
>
> Yes, we agree with you. We will definitely revise it.  We are going to include them in our revised manuscript. The fitting degree is to describe how deeply the model fits the training set. Typically, Cross Entropy loss forces the classifier to fit the training dataset with 0 loss. In Eq. 3, RelaxLoss sets a threshold to prevent the model from further fitting in the training set. The higher threshold means the loss function prevents the model from further fitting on the training set, and vice versa. This scenario provides a similar effect of Early Stopping technique which prevents overfitting and meanless fitting. In fact, the key to maintaining the testing accuracy is the third formulation in Eq. 3. When the model reaches the threshold, the third formulation in Eq. 3 plays an important role in maintaining the testing accuracy. By applying these points, we will refine the description of RelaxLoss according to your comments.
>
> > [3] It is unclear what Figure 1 is trying to tell us. There are brief descriptions in Sections 4.2 and 4.3. However, it would be beneficial to add more text explaining it, especially since it represents the core intuition of the work.
>
> Thank you for your suggestion. Yes, we will add more text to provide the core intuition of the work. Fig.1 displays that RelaxLoss can mitigate the distinguishability between the member and non-member distributions. That is because, if an attacker is more capable of differentiating the member/non-member distributions, the successful odds of attack go up. CenterLoss can improve the distinction of deep features to help the testing sample achieve better distribution. Actually, we have run new experiments and provided a new chart by adding CenterLoss chart - please click [HERE-NewPlots](https://anonymous.4open.science/r/UAI_CRL-269/CRL_UAI24Rebuttal.pdf). Our approach, CRL, can maintain the distinction while “relaxing” the model. Overall, we realized this point was not well stated in our original manuscript, so we will incorporate this in the revised manuscript according to your suggestion.
>
> > [4] It would be interesting to see CE with Center Loss in Figure 1.
>
> Thank you for your suggestion. According to your suggestion, (as we mentioned in the response above), we indeed have run experiments on it and provided a new chart - please click [HERE-NewPlots](https://anonymous.4open.science/r/UAI_CRL-269/CRL_UAI24Rebuttal.pdf). The result is consistent: both training and testing samples have a farther distance to the decision boundary, while the range of distance to the zero points becomes wider.
>
> > [5] Where is T_rcl used in equation 9?
>
> 𝜏_𝑟𝑐𝑙  is used to compute L_rcl, which is shown in Alg. 1 (Line 10-17) and Fig. 3.
>
> > [6] Understanding Figure 2 required a lot of back and forth between the text, equation 9, and Algorithm 1. Adding more description in the caption of Figure 2 and/or renaming "Scenario X" to something more meaningful may help.
>
> Thank you for your suggestion. Yes, we will add what we stated in the main text to the caption so that the readers can immediately and easily catch the information.
>
> > [7] The abstract mentions lack of consideration of deployment cost as a gap in the field. However, there is no evaluation of the deployment cost of CRL. Adding some results on cost will strengthen the paper.
>
> Thank you for your comments. According to your suggestion, we will include an evaluation of the deployment cost in the revised manuscript. The reason why we did not show the deployment cost is that the additional cost is insignificant. Both RelaxLoss and CenterLoss require no additional training resources and it has almost no impact on training speed, and so does CRL. Centers are included in the loss function part rather than in the Model itself. So, it never incurs any additional cost during inference. When the model is in evaluation phase, it acts as it typically does. In other words, no additional costs are incurred during evaluation.

---

### Official Review · Reviewer_5rTq · 2024-03-22

**Q2-1 Originality-Novelty:** 2
**Q2-2 Correctness-Technical Quality:** 3
**Q2-5 Clarity Of Writing:** 2

**Q1 Summary And Contributions:**

This paper proposes a new defense against membership inference attacks for classification models. The defense mainly involves a new loss function that aims to reduce the discrepancy between the model's output distributions of member and non-member data. Evaluations on three datasets and several models show that the defense can lower the attack's effectiveness.

**Q2-3 Extent To Which Claims Are Supported By Evidence:**

2: Fair: the main claims are somewhat supported by evidence (but the experimental evaluation may be weak, or does not match entirely with the claims, important baselines may be missing, proofs contain important ideas but lack rigor, algorithmic details are only discussed superficially, references are imprecise, assumptions are not sufficiently motivated or explicated, etc.).

**Q2-4 Reproducibility:**

3: Good: key resources (e.g. proofs, code, data) are available and key details (e.g. proofs, experimental setup) are sufficiently well-described for competent researchers to confidently reproduce the main results.

**Q3 Main Strengths:**

**Originality / Novelty**

* The proposed new loss function is novel to my understanding.

**Correctness / Technical Quality**

* The loss function, algorithms, and training steps look technically correct.
* Evaluations are generally comprehensive, including three datasets, several models, and several defenses.

**Claims Supported**

* The claim that the proposed can outperform other defenses is well-supported.

**Reproducibility**

* Most experimental and training details are provided.

**Clarity**

N/A

**Q4 Main Weakness:**

**Originality / Novelty**

* I don't have major issues with the novelty. The technique generally makes sense to me.

**Correctness / Technical Quality**

* A minor point is that the proposed defense feels more like a loss function than a new training paradigm. It still relies on the common supervised training techniques.
* **[Q1]** The validity of Figure 1 needs more justifications. Can you explain why the two distances are important and are the right metric to check if two distributions are overlapping? Besides, the two distributions are still not that overlapping in CRL (or still hard to tell). This seems to be the only metric that measures the discrepancy between the two distributions in this paper. Is it possible that some attacks are able to identify the discrepancy between two distributions, where the difference is beyond what these two distances could measure?

**Claims Supported**

* **[Q2]** It seems that the following claimed observation is not demonstrated or evaluated: "we observed that the privacy vulnerability of the model is closely correlated with the gap between the model's data-memorizing ability and generalization ability." Are there any results or references to support this claim?

**Reproducibility**

N/A

**Clarity**

* **[Q3]** The overall motivation of the methodology is unclear. After reading the paper, it is still unclear to me (1) what the three subsections in Section 4 are presenting and (2) how these losses can "enhance and maintain models' prediction confidence in non-member data  while mitigating overconfidence the models' prediction distribution on member data." I think the main problem here is that, all techniques in Section 4 are brought up out of the blue. For example, why should we care about RelaxLoss in Section 4.1? There is no context for what loss is designed and why it is so. The overview (Algorithm 1 and Figure 3) was not explained until the end of all details. Algorithm 1 was also never motivated.
* A minor point that $f$ was overloaded in Section 3, first as a target model and then as the shadow model.

**Q5 Detailed Comments To The Authors:**

See the 3 questions above.

**Q9 Complying With Reviewing Instructions:**

Yes

---

> ### Author Rebuttal · Authors · 2024-04-08
>
> We thank the reviewer for the constructive comments. We hope that we were able to address all your concerns. We sincerely hope the responses help you consider raising your score.
>
> > [Q1]
>
> Thanks for your questions. We address the concerns below:
>
> (1). The distance to the decision boundary is directly related to the prediction distributions. It can be regarded as a metric to measure how close two prediction distributions are (shown in Fig.6). The distance to the zero point is a trait that leads to the disparity of the distance to the decision boundary. In other words, it indirectly affects the prediction distributions. Our experiments in Fig. 1 show that the alignment in the distance to the zero point can help the alignment in prediction distribution. By the way, in the original Fig 1, it was hard to recognize the overlap because of the color tones of 20K+ samples. Hence, we re-plotted them [HERE-NewPlots](https://anonymous.4open.science/r/UAI_CRL-269/CRL_UAI24Rebuttal.pdf) for better readability.
>
> (2). All charts in Fig. 1 show the scenario which the model loses little or no testing accuracy. With more strict hyper-parameters, they can overlap more and more. As for MIAs, all MIAs are trying to detect the behavioral disparity of the model. It is impossible to identify the disparity from the original inputs if these two distance distributions are fully aligned.
>
> > [Q2]
>
> Yes, we explain the data-memorizing ability as that different models have different fitting abilities on the training set. The ResNet’s data-memorizing ability is higher than VGG. That is why ResNet is easier to attack. The gap between them means the discrepant prediction distribution, accuracy, and representation. These disparities are shown Fig.1 and Fig.6. It can be seen that in both Fig.1 and Fig.6 the model trained with cross-entropy has a clear gap between training and testing distributions. The way we compute the distance to the decision boundary is by using the 1st largest confidence minus the 2nd largest confidence. It naturally reflects the prediction discrepancy between training and testing samples. Hence, when they get more aligned, the privacy risk can be better mitigated.
>
>
> > [Q3]
>
> We are happy to explain it.
> (1). Sec 4.1 and Sec 4.2 introduce RelaxLoss and CenterLoss as background knowledge since their insights help understand our approach presented in Sec 4.3. RelaxLoss is to mitigate MIAs by reducing distinguishablitiy between the member and non-member loss distributions. CenterLoss is to enhance the discriminative power of the deeply learned features. Our motivation is to keep the discriminative power of deep features while “relaxing” the model.
>
> (2). “why should we care about RelaxLoss in Section 4.1?”: we care about it because it shows effective performance without additional cost. In Sec 4.1, we introduce RelaxLoss. In Fig.1, we explore what RelaxLoss is not capable of and understand that there is room for RelaxLoss to improve. Then, we introduce CenterLoss that improves Cross-Entropy to mitigate the issue. Finally, in Sec. 4.3., we propose CRL which we build on top of the advantages of Improved RelaxLoss and Relaxed Center Loss, but we advance CRL with our novel observation, which is devised from the discrepancy of training and testing distributions.
>
> As the reviewer pointed out, we realized that the connection between Sec, 4.1-4.2 and Sec 4.3 was not smoothly stated. Hence, we will revise the manuscript to fix this, as we addressed above. Also, in the revised version, we will motivate and provide an overview of Algorithm 1 and Figure 3 early on to incorporate your comment.

---

### Official Review · Reviewer_pYNE · 2024-03-25

**Q2-1 Originality-Novelty:** 2
**Q2-2 Correctness-Technical Quality:** 3
**Q2-5 Clarity Of Writing:** 3

**Q1 Summary And Contributions:**

The authors propose a combination of two losses --- Relax loss and Center loss, to mitigate membership inference attacks. In particular, a) they introduce temperature parameters in the CE loss term of relaxed loss, and b) they use center loss to force the points of the same class to be close in representation space. The authors propose an improved relax loss variant that uses a temperature higher than 1 in the softmax loss. The main innovation is the idea to avoid overfitting by forcing the model representations not be too close to each other and similarly logits to be not too high. This is achieved by reversing the sign of the loss in every even epoch if the loss is below a predecided threshold. The authors evaluate 3 models on 3 vision datasets and show that the models trained with their approach achieve better test accuracy while having similar or slightly lower price leakage.

**Q2-3 Extent To Which Claims Are Supported By Evidence:**

3: Good: the main claims are supported by convincing evidence (in the form of adequate experimental evaluation, proofs, (pseudo-)code, references, assumptions).

**Q2-4 Reproducibility:**

3: Good: key resources (e.g. proofs, code, data) are available and key details (e.g. proofs, experimental setup) are sufficiently well-described for competent researchers to confidently reproduce the main results.

**Q3 Main Strengths:**

- The method is straightforward and easier to implement.
- The empirical results are promising. In particular, the method performs better (test accuracy) in cases with comparable or lower privacy leakage. I appreciate that the authors reported results with several attack techniques.

**Q4 Main Weakness:**

- The approach combines two existing with little innovation or insight as to why one can't work. The ablation results justify their choices, but a rationale needs to be included.
- Some claims are unsupported. For example, the abstract claims, "...we observed that the privacy vulnerability of the model is closely correlated with the gap between the model’s data-memorizing ability and generalization ability." However, no analysis has been done to support this claim.
- The writing could be improved. Some examples and questions are below:
    - In Related Work:
         - .. to training shadow models ... -> ... to train shadow models ...
         - .. some studies against ...  -> .. some studies to defend against ...
    - Section 3:
         - "Developing an attack model, it needs to mimic the target models’ prediction distribution." -> This phrase is weird; please review.
    - 4.1 Mechanism on Relaxloss -> Mechnism of Relaxloss (similarly sec 4.2)
    - What is "fitting degree" in Sec 4.1?
    - What does increasing random seed mean in 5.1 (Datasets)

**Q5 Detailed Comments To The Authors:**

- What is the relevance of distance to 0? Why does it concern membership inference attacks? This was unclear to me.

- Fig 1. Why is the distance to the decision boundary clipped around 1? It is also unclear what role distance from origin plays in membership inference attacks.

- Why are attack accuracies of Grad lower? White box attacks have more information to launch attacks and hence should be more accurate/effective than other black box attacks.

- The method tempers probability/logit vectors to avoid overfitting. However, some recent works have shown that membership inference attacks can happen even without overfitting [1]. It is relevant to evaluate the method against such attacks.

- Given the metrics are very close to each other, I think you should report the variance. A variance of up to 3.5% could be significant for these results.

[1] Li, Zheng, and Yang Zhang. "Membership leakage in label-only exposures." Proceedings of the 2021 ACM SIGSAC Conference on Computer and Communications Security. 2021.
[2] Choquette-Choo, C.A., Tramer, F., Carlini, N. &amp; Papernot, N.. (2021). Label-Only Membership Inference Attacks. <i>Proceedings of the 38th International Conference on Machine Learning</i>, in <i>Proceedings of Machine Learning Research</i> 139:1964-1974 Available from https://proceedings.mlr.press/v139/choquette-choo21a.html.

**Q9 Complying With Reviewing Instructions:**

Yes

---

> ### Author Rebuttal · Authors · 2024-04-08
>
> **Rebuttals by Authors (1/2)**
>
> We thank the reviewer for the thoughtful comments. We hope that we were able to address all your concerns. We sincerely hope the responses help you consider raising your score.
>
> **Q4 Main Weakness:**
> > The approach combines two existing with little innovation or insight as to why one can't work. The ablation results justify their choices, but a rationale needs to be included.
>
> Thank you for your comments, RelaxLoss is to mitigate MIAs by reducing distinguishability between the member and non-member loss distributions. CenterLoss is to enhance the discriminative power of the deeply learned features. Our motivation is to keep the discriminative power of deep features while “relaxing” the model. In Fig.1, we explore what RelaxLoss is not capable of and understand that there is room for RelaxLoss to improve. Then, we introduce CenterLoss which improves Cross-Entropy to mitigate the issue. Finally, in Sec. 4.3., we propose CRL because just by combining the two the goal cannot be achieved. We build CRL on top of the advantages of Improved RelaxLoss and Relaxed Center Loss, but we advance CRL with our novel observation, which is devised from the discrepancy of training and testing distributions. That is, we utilize the "relaxation" mechanism to make the training distribution closer to the testing distribution and use the centralization mechanism to prevent the degradation of the model on the testing distribution so that the two distributions more overlap. Thanks to your comment, we will incorporate this explanation in the revised manuscript.
>
> > Some claims are unsupported. For example, the abstract claims, "...we observed that the privacy vulnerability of the model is closely correlated with the gap between the model’s data-memorizing ability and generalization ability." However, no analysis has been done to support this claim.
>
> We are happy to explain it. we mean by the data-memorizing ability that different models have different fitting abilities on the training set. The ResNet’s data-memorizing ability is higher than VGG. That is why ResNet is easier to attack. The gap between them means the discrepant prediction distribution, accuracy, and representation. These disparities are shown Fig.1 and Fig.6. It can be seen that in both Fig.1 and Fig.6 the model trained with cross-entropy has a clear gap between training and testing distributions. The way we compute the distance to the decision boundary is by using the 1st largest confidence minus the 2nd largest confidence. It naturally reflects the prediction discrepancy between training and testing samples. Hence, when they get more aligned, the privacy risk can be better mitigated. Thank you for pointing it out, and we will definitely include this explanation in the revised manuscript.
>
>
> > The writing could be improved. Some examples and questions are below:
> > In Related Work:
> > .. to training shadow models ... -> ... to train shadow models ... ;
> > .. some studies against ... -> .. some studies to defend against …
>
> Thank you for pointing them out. Sure, we will update them accordingly.
>
> > Section 3:
> "Developing an attack model, it needs to mimic the target models’ prediction distribution." -> This phrase is weird; please review.
>
> Apologies for the confusion. We agree that the phrase was not well written. We will rephrase it as “If one were to develop an attack model, the model needs to mimic the target model's prediction distribution” and update it in the revised manuscript.
>
> > 4.1 Mechanism on Relaxloss -> Mechnism of Relaxloss (similarly sec 4.2)
>
> We will fix it in the revised manuscript.
>
> > What is "fitting degree" in Sec 4.1?
>
> The fitting degree is to describe how deeply the model fits the training set. Typically, Cross Entropy loss forces the classifier to fit the training dataset with 0 loss. As shown in Eq. 3, RelaxLoss sets a threshold to prevent the model from fully fitting in the training set.
>
> > What does increasing random seed mean in 5.1 (Datasets)
>
> We are happy to answer. It is a data-splitting strategy. The training datasets of the shadow models need to partially overlap each other due to the limited number of samples. To achieve this, when we split the data for the i-th shadow training set, we set the random seed to x+i (x is the base seed).

---

### Meta-Review · Area_Chair_a1H8 · 2024-04-18

This paper proposes an approach for defense against membership inference attacks for models trained for classification tasks. The authors exploit two losses in their methodology: RelaxLoss (borrowed from Chen et al.) and a center loss. In particular, they use center loss to enforce the representation of each class being similar in the representation space. This property will help mitigate the vulnerability of the model against membership inference attacks. The proposed method is evaluated on extensive experiments on classification tasks.

While there are concerns about the presentation, the reviewers found the proposed loss function novel and the experiments detailed enough. The ideas in the paper can have a positive impact on the privacy community in general. I encourage the authors to address the reviewers' comments about the edits in the presentation.